# Diverse Evolution in 111 Plant Genomes Reveals Purifying and Dosage Balancing Selection Models for *F-Box* Genes

**DOI:** 10.3390/ijms22020871

**Published:** 2021-01-16

**Authors:** Zhihua Hua

**Affiliations:** Interdisciplinary Program in Molecular and Cellular Biology, Department of Environmental and Plant Biology, Ohio University, Athens, OH 45701, USA; hua@ohio.edu; Tel.: +1-740-593-1123

**Keywords:** F-box, SCF ubiquitin ligases, evolution, purifying selection, balancing selection, yeast two-hybrid, protein-protein interactions, plants

## Abstract

The F-box proteins function as substrate receptors to determine the specificity of Skp1-Cul1-F-box ubiquitin ligases. Genomic studies revealed large and diverse sizes of the *F-box* gene superfamily across plant species. Our previous studies suggested that the plant *F-box* gene superfamily is under genomic drift evolution promoted by epigenomic programming. However, how the size of the superfamily drifts across plant genomes is currently unknown. Through a large-scale genomic and phylogenetic comparison of the *F-box* gene superfamily covering 110 green plants and one red algal species, I discovered four distinct groups of plant *F-box* genes with diverse evolutionary processes. While the members in Clusters 1 and 2 are species/lineage-specific, those in Clusters 3 and 4 are present in over 46 plant genomes. Statistical modeling suggests that *F-box* genes from the former two groups are skewed toward fewer species and more paralogs compared to those of the latter two groups whose presence frequency and sizes in plant genomes follow a random statistical model. The enrichment of known Arabidopsis *F-box* genes in Clusters 3 and 4, along with comprehensive biochemical evidence showing that Arabidopsis members in Cluster 4 interact with the Arabidopsis Skp1-like 1 (ASK1), demonstrates over-representation of active *F-box* genes in these two groups. Collectively, I propose purifying and dosage balancing selection models to explain the lineage/species-specific duplications and expansions of *F-box* genes in plant genomes. The purifying selection model suggests that most, if not all, lineage/species-specific *F-box* genes are detrimental and are thus kept at low frequencies in plant genomes.

## 1. Introduction

Genome innovation is largely attributed to the rapid birth-and-death of gene family members, particularly those encoding the large cohort of protein workhorses in cells [1]. The advancement of genome sequencing has uncovered an enormous number of protein families. For example, since the first version of the Pfam protein family database was established in 1995 [2], its size has risen from 1000 to 18,259 families in the current version (V33.1, as of May 2020, https://pfam.xfam.org), reflecting complex proteomic systems that control diverse life processes.

In addition to large size, dynamic evolutionary changes of protein families across the kingdoms of life have resulted in numerous conserved and specialized regulatory pathways at different phylogenetic levels. For example, the ubiquitin (Ub)-26S proteasome system (UPS) is designed to degrade a myriad group of short-lived or abnormal proteins in all eukaryotic organisms [3,4,5]. This system generally involves the sequential action of Ub-activating (E1), Ub-conjugating (E2), and Ub-ligating (E3) enzymes that mediate the ubiquitylation of substrates that are subsequently degraded by the 26S proteasome [4,5,6,7,8]. Although the process of ubiquitylation and degradation is conserved across all eukaryotes, the families of proteins involved in the UPS have dramatically expanded in multicellular organisms compared to those in the unicellular life system. Even among multicellular organisms, the sizes of these families vary significantly [9]. One common hypothesis believes that such diverse changes in UPS family sizes and types reflect the adaptive role(s) of the UPS in proteomic regulation [9,10]. Supporting this idea, the extremely large expansion of the UPS in land plants has been argued to benefit their sessile lifestyle [7,11,12,13].

The F-box (FBX) protein family is an extraordinary example demonstrating the large expansion of a UPS family in the plant kingdom [14,15,16,17]. *FBX* genes that have been characterized in the model plant *Arabidopsis thaliana* (Arabidopsis hereafter unless otherwise described) are involved in virtually all aspects of plant growth and development, including seed germination [18,19], photomorphogenesis [20,21,22,23], circadian rhythms (see review by [24]), cell cycle [25,26,27], epigenetic regulation [28], stress responses [29,30], hormone signaling (see reviews by [7,31]), floral organ establishment [23,32,33], self-incompatibility [34,35,36], and embryogenesis/seed development [23,37]. Since the first genome of Arabidopsis was sequenced in its reference accession, Columbia-0 (Col-0), in 2000 [38], 83 *F-box* genes have been genetically characterized to date (as of 1 October 2020) [39]. However, compared to the predicted ~800 *FBX* genes in Arabidopsis, the fraction of functionally characterized *FBX* members is much lower than many angiosperm core gene families [7,40].

In attempting to address the dilemma between the activity and size of the *FBX* gene superfamily in Arabidopsis, we previously compared the phylogeny of the *FBX* superfamily across 18 plant genomes, ranging from one algal species and two basal land plants to four monocotyles and eleven eudicotyles [15]. We arbitrarily classified the plant *FBX* genes into small and large taxonomic scale protein (STSP and LTSP) coding gene groups based on the number of genomes in which an *FBX* gene was present (four or fewer genomes for STSP, and greater than 4 out of the 18 plant genomes for LTSP). Such an arbitrary classification allowed us to discover significant enrichment of active *FBX* genes in the LTSP group compared to the STSP group. Through epigenomic comparisons, we further concluded that epigenetic suppression of gene expression contributed to the functional differentiation between LTSP and STSP *FBX* genes in Arabidopsis [41].

Although we do not yet know whether this epigenetic suppression of gene expression is a cause or a consequence of the large expansion of the STSP *FBX* gene group in plants, we recently discovered diversifying evolutionary processes within UPS gene families in two phylogenetically distantly related plant species, Arabidopsis and rice [10,42]. First, we identified that expression of the UPS founding member, *Ub*, is differentially regulated in Arabidopsis and rice due to a putative retrotransposition duplication mechanism in propagating the *Ub* moieties of the *poly-Ub* genes. Although both plants shared 100% identical Ub amino acid sequences, the rice *UBQ2* promoter drives the expression of a *poly-(Ub)_6_* gene much less effectively than the Arabidopsis *UBQ10* promoter in Arabidopsis seedlings, suggesting that these two most highly expressed *UBQ* genes in each species are regulated in a species- or lineage-specific manner [42]. Second, through a comprehensive genomic comparison of 11 UPS protein families between *Oryza* and *Arabidopsis* genera, we further detected distinct evolutionary and duplication patterns of UPS protein families. For example, a greater number of *Arabidopsis FBX* genes are under pseudogenization and neutral changes than *Oryza FBX* genes, which raised a new question about the contribution of epigenomic regulation in shaping the functional divergence of *FBX* genes in rice. As the rice genome encodes an even larger number of *FBX* genes than Arabidopsis [10,14,15], the smaller fraction of rice *FBX* genes under pseudogenization or neutral changes suggested that distinct evolutionary processes are occurring for the *FBX* genes of these two distantly related species.

Gene duplications in a genome can result from whole genome duplications (WGDs), segmental inter- or intra-chromosomal duplications, and small-scale duplications (SSDs), which can be further divided into tandem and retrotransposition duplications [43,44,45]. The consequences of these duplication mechanisms significantly vary. Gene dosage-balance constraints have been hypothesized to guard duplicated gene copies against loss by determining the stoichiometry of duplicated products [46]. While the dosage of WGD-mediated gene duplicates can be relatively easy to preserve, the stoichiometry of partners involved in the same biochemical pathway can frequently be disrupted if only one or a few are duplicated through SSDs, leading to harmful evolutionary outcomes that are thus unlikely to be maintained.

The gene dosage-balance hypothesis assumes that all gene duplicates remain active following their duplication. However, this is not always the case. For example, many retrotransposed genes are not fused with a promoter, thus rendering them inactive [47]. We also discovered that a large fraction of SSD-produced STSP *FBX* genes are epigenetically silenced by RNA-directed DNA methylation and histone H3K27 trimethylation [41]. We hypothesized that transcriptional inactivation of *FBX* duplicates through epigenomic programming is therefore a potential evolutionary mechanism to maintain proteome stability [41]. According to this possibility, the retention of duplicate members of a gene family in a genome may not only be attributed to gene dosage-balance among partners involved in the same metabolic pathway(s), but also to gene inactivation. Although the latter process may lead the gene duplicates to pseudogenize more rapidly due to neutral changes, their fast duplication could result in the large size variance of the *FBX* gene family even between closely related plant species [15,41].

Given the vast biochemical roles and potentially diverse evolutionary processes within the plant *FBX* gene superfamily, it is critical to understand which plant *FBX* genes are active or inactive, and their phylogenetic distributions across different plant lineages. Here, I carried out a large-scale comparative genomic study of the *FBX* gene superfamily across 110 green plant species, plus one red alga as an outgroup. Using an unsupervised k-means clustering approach, I discovered four groups of green plant *FBX* genes that are under distinct duplication retention rates, functional constraints, and phylogenetic distributions. This study not only provides genomic evidence supporting the diversifying evolutionary process of the *FBX* gene superfamily across distant plant lineages, but also sheds light on the function of genome duplications. In addition, the resulting dataset provides an evolutionary basis for guiding future functional genomic studies of the *FBX* genes across a wide phylogenetic range of plant genomes.

## 2. Results

### 2.1. Deep Annotation of FBX Genes in a Large Set of Plant Genomes

An FBX protein is defined by the presence of an N-terminal ~60 amino-acid FBX domain (FBXD) that is linked to the Cul1 scaffold protein via interaction with an Skp1 protein, in the so-called Skp1-Cul1-*FBX* (SCF) ubiquitin ligase complex [7,39,48]. Therefore, identification of an FBXD sequence has been used as an in silico standard to predict a putative FBX protein. HMMER searches using a Pfam-HMM profile built upon a seed set of known FBXD sequences have been widely used as an effective way to identify FBXD sequences [14,15,17,49]. However, identifying the complete set of *FBX* genes encoded in a genome is not trivial, requiring the searching of both annotated and unannotated genes. Only if lists of *FBX* genes are complete or equally annotated will comparisons across genomes be close to the real distribution pattern of *FBX* genes in plants. Moreover, to increase the statistical power of cross-genome comparisons, a large set of genomes is also necessary. To reach this goal, I applied a recently developed bioinformatic program, called Closing-Target-Trimming (CTT) [50], to identify a nearly complete set of *FBX* genes in 111 plant species by both searching annotated *FBX* genes from each proteome file and reannotating new *FBX* loci in each genome dataset available at Phytozome V13 (Appendix A).

In total, CTT reannotation used 14,910 nonredundant FBXD sequences (identities ≤70%), including 14,245 sequences predicted by the HMMER-Pfam search from 111 previously annotated plant proteomes and 665 seed sequences retrieved from the Pfam database, as queries to tBLASTn search each genome to find putatively unidentified *FBX* loci. After trimming away known linked *FBX* loci, 153,172 out of 163,868 unique tBLASTn hits were annotated by GENEWISE [51], which used the best BLASTx hit from 68,751 prior annotated FBX protein sequences as a reference. The 57,048 peptide sequences predicted by GENEWISE were again subject to a HMMER-Pfam search to identify in total 9720 putative FBXD-containing protein sequences. The final predicted *FBX* protein and coding sequences in the 111 plant genomes are included in Appendix A, respectively. The remaining 27,707 loci predicted by GENEWISE to contain a frame shift or premature stop codon were considered as *FBX* pseudogenes in each genome (Figure 1A and Appendix A). 

Count assays show that the number distribution of new *FBX* loci has a relatively good correlation with that of prior annotated *FBX* genes in many genomes (Figure 1B and Appendix A). However, in some genomes, like *Hordeum vulgare* (*Hvul*), *Helianthus annuus* (*Hann*), *Triticum aestivum* (*Taes*), *Daucus carota* (*Dcar*), and *Miscanthus sinensis* (*Msin*), more putative *FBX* loci were identified (Figure 1B,C), which reflects unequal annotation qualities among the genomes available at Phytozome V13. Although the number of pseudogenes may be over-estimated because the presence of an *FBXD* in each locus is unknown, they show the strongest correlation with the count of new *FBX* loci and genome sizes (Figure 1B,C and Appendix A). Therefore, the birth and death of an *FBX* locus is rapid in plants. One mechanism could be attributed to the high rate of polyploidization events [43,45]. Notably, more pseudogenes than *FBX* protein-coding genes were identified in five out of 111 species*,* including three Poaceae species (*Hvul*, *Msin*, and *Zea mays* (*Zmay*)), one Superasterid (*Hann*), and one Fabales (*Arachis hypogaea* (*Ahyp*)) (Figure 1C). Rapid pseudogenization could be an evolutionary mechanism to keep gene dosage balance in these species; for example, *Zmay* only encodes half the number of *FBX* loci as *Sorghum bicolor* (*Sbic*), but has 5.7-fold more putative pseudogenes than are identified in *Sbic* (Figure 1C). As *Zmay* and *Sbic* are evolutionarily closely related and were split only 12 million years ago (mya) [52], the larger number of pseudogenes and the smaller number of *FBX* loci identified in *Zmay* compared to *Sbic* suggest that the number of active or functioning *FBX* genes could be smaller than what is predicted from the genomes.

The mild correlation of number of *FBX* loci (including new and prior annotated ones) with genome size indicates that genome amplification is a driving force contributing to the expansion of the *FBX* superfamily in plants (Appendix A). However, this driving force is not likely the primary determinant, because the size of the *FBX* family can vary significantly even between phylogenetically closely related species. In addition to the striking size difference of the *FBX* families in *Zmay* and *Sbic* described above, such variation is also evident in the four Rosaceae species: *Cucumis sativa* (*Csat*), *Fragaria vesca* (*Fves*), *Prunus persica* (*Pper*), and *Malus domestica* (*Mdom*). While the first three species have a genome around 200 Mb and the fourth one has a 710 Mb genome due to a recent WGD [53], their *FBX* families were identified to contain 204, 939, 468, and 621 members, respectively. The *FBX* family in *Fves* is 2.0- and 4.6-fold larger than those of *Pper* and *Csat*, respectively, although their genome sizes are nearly equal (Figure 1C). However, the greater number of *FBX* genes identified in *Mdom* than in *Csat* and *Pper* suggests that WGDs could play a role in increasing family size.

### 2.2. Clustering the Green Plant FBX Genes

Our previous studies identifying the role of epigenetic suppression of gene expression in a large set of Arabidopsis STSP *FBX* genes, and the finding here that the *Zmay FBX* family has 4.9-fold more putative pseudogenes than the protein coding genes, collectively argue that plant *FBX* genes have different levels of activity (Figure 1C; [41]). Given that many silenced Arabidopsis STSP *FBX* genes are SSDs, lineage-specific, and recently duplicated [15], I hypothesized that such differential activities can be clustered based on their evolutionary relationship. To rank this relationship, I developed three criteria as follows: (1) The more species the *FBX* orthologous group has, the more active its *FBX* genes are; (2) the further the phylogenetic distance of two species within an orthologous group, the more conserved the *FBX* genes are; and (3) the fewer species from which an *FBX* gene has been lost between two distant species of an orthologous group, the more active the *FBX* genes are.

OrthoMCL has been widely used to find orthologous groups of a gene superfamily [54]. This analysis also provided relatively accurate orthologous relationships between *FBX* genes. For example, orthoMCL0000 identifies exactly four members of the TIR1/AFB1-3 subfamily (Appendix A; [55]). Similarly, orthoMCL0005 contains two Arabidopsis *FBX* genes that encode EBF1/2 known for targeting EIN3 for ubiquitylation during ethylene signaling (Appendix A; [56,57,58]). Therefore, I considered each OrthoMCL group as an *FBX* subfamily. In total, 5858 *FBX* subfamilies were identified.

To quantify the activity of *FBX* genes, I counted the number of species (cnt), summed up between species phylogenetic distance (dist), and calculated the number of missing species between the two distant species (gap) in each subfamily. Apparently, a large group of *FBX* subfamilies are lineage-specific (Figure 2A); the sum of between-species phylogenetic distance separated the subfamilies better than the number of species in lineage-specific subfamilies, although the two values show an overall strong correlation (Figure 2B,C; Spearman’s ρ = 0.93, *p* < 2.2 × 10^−16^). Additionally, the number of subfamilies exponentially declined upon an increase in the number of missing species (Figure 2D).

Applying a resampling strategy in combination with k-means clustering on this dataset [59], I found four well-separated clusters of *FBX* subfamilies (Figure 3A,B). When I assessed the cnt density distribution of the entire set of *FBX* subfamilies, a bimodal-shaped curve could be identified, revealing a large fraction of *FBX* genes possessing low cnt values, and a small fraction peaking at high cnt values, with a flat curve present in between (Figure 3C). Such a distribution is similar to previous findings in the *FBX* genes of 18 plants and a large set of 69,133 angiosperm gene families in 36 flowering plants [15,40]. Interestingly, the four clusters can be ordered sequentially from low cnt values to high cnt values, suggesting that they were differentiated primarily by species- or lineage-specificities (Figure 4A–D). Further analysis of the cnt density distribution in each cluster identified uniform and Weibull distributions in Clusters 3 and 4, respectively, but no statistical models that could fit the cnt changes across the entire set of *FBX* subfamilies, nor those in Clusters 1 and 2 or a combination of the first and the latter two clusters (Figure 3C and Figure 4, Appendix A). Therefore, the four clusters of *FBX* genes appear to be under differential evolutionary histories.

The biased distribution of *FBX* subfamilies in Clusters 1 and 2 toward low cnt values indicates a strong selection that suppresses the spread of most *FBX* genes across plant species. While this could support lineage-specific roles of *FBX* subfamilies in Clusters 1 and 2, the proportion of *FBX* subfamilies in Cluster 1 is dramatically higher than the fraction of lineage-specific gene families identified in angiosperms, if we consider Cluster 4 as a core *FBX* gene group (Fisher’s exact test, *p* = 2.6 × 10^−137^; Appendix A, [40]). However, this comparison is likely an underestimation, because the lineage-specific angiosperm gene families include the as-yet-unidentified Cluster 2 and Cluster 3 groups. Therefore, the birth of many lineage-specific *FBX* genes is not necessary to be functionally relevant but rather a consequence of a strong selection against their fixation in many genomes.

### 2.3. Selective and Random Birth of FBX Genes

The well-fitted uniform and Weibull distribution patterns of cnt values of *FBX* subfamilies from Clusters 3 and 4 suggested that their presence and absence in plant genomes resulted from stochastic events. However, the distribution of *FBX* subfamilies from Clusters 1 and 2 in plants appears to be under strong selection due to their skewed presence toward fewer species (Figure 4). I therefore asked whether their numbers also significantly differ across plants. I plotted the number of *FBX* genes per genome from each cluster, as well as the full set of *FBX* subfamilies and the remaining orphan loci, across 111 plant species, according to their phylogenetic order (Figure 5). I surprisingly found an overall similar *S* curve distribution regarding the number of *FBX* genes across the plant kingdom from the full set of subfamilies, the orphan group, and Cluster 1, which peaked in Brassicales and Poceae species and fell in non-Brassicale rosids, basal embryophytes, and green algae (Figure 5A–C and Appendix A). Consequently, these three groups of *FBX* genes demonstrate a strong number correction among the 111 plant genomes (Appendix A). The distribution of Cluster 2 *FBX* genes is unique, being significantly enriched in Brassicales and depleted in the other plant genomes (Figure 5D). Clusters 3 and 4 show the third distribution pattern, in which non-Brassicale rosids encode the highest number of *FBX* genes on average (Figure 5E,F).

To estimate the birth and death patterns of *FBX* genes among these groups, I modeled the density distribution of the total number of *FBX* genes in each species with or without the orphan genes to a Weibull function (Appendix A), suggestive of an overall random birth and death evolution. Those from Clusters 1 and 3 fit well to a gamma distribution, and that from Cluster 4 fits a lognormal distribution (Figure 6). However, no distribution model could be found to fit the density of *FBX* gene numbers in Cluster 2. It was unexpected to find out that no model could fit the number distribution of orphan genes across plants (Appendix A), but collectively, this analysis further suggests the presence of both selective and random birth and death patterns in different plant *FBX* gene groups.

### 2.4. Differential Retention Rates and Functional Constraints among Four FBX Clusters

Gene duplications followed by functional diversification have been recognized as a common theme in contributing to the expansion of gene superfamilies [43]. While high rates of WGDs and SSDs could explain the dramatic expansion of UPS families in flowering plants, the retention rates of duplicates were not homogeneous across different groups of genes [40], meaning that both types and retention rates of gene duplications could result in the selective and random birth and death patterns in the four different plant *FBX* gene clusters. To test this hypothesis, I analyzed retention rates of WGD paralogs, compared fractions of WGDs and SSDs, and predicted proportions of neutrally changing paralogs in the four clusters of *FBX* genes. 

Due to various chromosomal rearrangements following WGDs, it is challenging to determine the members in a gene superfamily that arose via WGDs. To address this, Li et al. (2016) applied two approaches to study WGD events in 9178 core angiosperm gene families [40]. The first is based on gene tree and species tree reconciliation, while the second applies a Gaussian mixture modeling (GMM) approach of *K_s_* (number of synonymous substitutions per synonymous site)-based age distributions to predict duplication time. Due to the difficulty in reconstituting the gene tree of 78,471 *FBX* genes in 111 plant species, and the effectiveness of the GMM approach in modeling duplicate retention dynamics over time, I adopted the *K_s_*-based age distribution data of 27 plant species that were also studied by Li et al. [40] and assigned the *FBX* duplicates as the products of either SSDs and recent (<50 mya), K-Pg (Cretaceous-Paleogene) boundary (~50–70 mya), or ancient (>75 mya) WGDs.

Similar to the study of core angiosperm gene families, fractions of *FBX* WGD duplicates decline upon age in an L-shaped curve that can also be estimated as a power-law function (Appendix A, [40]). Considering that WGDs often produce active gene members [43], the quick reduction of *FBX* genes resulting from the early WGD events in 27 plants suggests that the rapid expansion of active *FBX* genes is harmful. If this is true, I hypothesize that the large expansion of the *FBX* gene family in plants is attributed to SSDs, and that many of them are inactive. To examine this, I compared the fractions of WGDs and SSDs among four clusters of *FBX* genes in 27 plant genomes. Consistently, fractions of WGD *FBX* members decline much more rapidly in Cluster 1 than in the other Clusters, and these are biased toward SSDs (Figure 7A–C and Appendix A, *p* < 0.05 for all comparisons; Kruskal–Wallis test followed by Dunn’s test with Benjamini–Hochberg multiple testing correction). Although the dynamics of WGD retention rates do not differ significantly among the remaining three clusters (Figure 7A and Appendix A), their SSDs are differentially represented. Cluster 4 is enriched and depleted in WGDs and SSDs, respectively, while Cluster 3 contains approximately equal fractions of *FBX* genes duplicated through the four types of duplications analyzed. Cluster 2, similarly to Cluster 1, has more fractions of SSD and K-Pg boundary duplicates than duplications originating from ancient and recent WGD events (Figure 7D). 

Compared to the full set of duplicated *FBX* genes, SSD duplications are over-represented in Cluster 1 and under-represented in the other three clusters (*p* = 3.7 × 10^−73^, 1.7 × 10^−2^, 5.0 × 10^−6^, and 4.3 × 10^−199^ for Clusters 1, 2, 3, and 4, respectively; Fisher’s exact test with Bonferroni multiple testing correction), confirming that a large proportion of plant *FBX* genes were duplicated through SSDs. While duplicates from recent and ancient WGDs are over-represented in Clusters 3 and 4, those amplified during the K-Pg boundary are enriched in Cluster 2 and under-represented in Clusters 1 and 3 (Figure 7D). Such biased duplication patterns resulted in differential functional constraints for *FBX* genes in the four clusters. For example, compared to the full set of duplicated *FBX* genes, neutrally evolving members (*K_a_*/*K_s_* ~ 1, the ratio of the number of nonsynonymous substitutions per nonsynonymous site (*K_a_*) to the number of synonymous substitutions per synonymous site (*K_s_*)) are under-represented in Cluster 4 in all four types of duplications examined, whereas they are over-represented in Cluster 1 in all WGD events and in Cluster 2 if they were duplicated from recent WGDs (Figure 7E). 

### 2.5. Diversifying Evolution of FBX Genes in Six Groups of Green Plants

The selective and random birth and death of *FBX* genes in the four clusters suggest diversifying evolution. To test this model, I clustered the 111 plants analyzed in this work based on the proportion of *FBX* genes from the four clusters and the orphan gene group in each species (Figure 8A and Appendix A). Interestingly, such clustering matches the phylogenetic relationship of the 111 plants, resulting in six relatively well-separated groups, particularly the groups of Brassicales and Poaceae plants, in which we have previously discovered diversifying evolutionary patterns within UPS families [10]. While Brassicales are enriched with *FBX* genes from Clusters 1 and 2, followed by Cluster 4 and orphan genes, Poaceae species have significantly more Cluster 1 *FBX* genes than Brassicales but have depleted the *FBX* gene members from Clusters 2 and 3 and slightly reduced the number of *FBX* genes from Cluster 4 and the orphan gene group. Although rosid plants experienced a common paleohexaploidization event [60], non-Brassicale rosids retain more *FBX* genes from Clusters 3 and 4 and fewer *FBX* genes from Cluster 2 and the orphan gene groups than the Brassicales. The distribution of *FBX* genes in basal embryophytes is similar to that in non-Brassicale rosids, except for a reduction in Cluster 3 members. The remaining non-rosid eudicotyle plants form a fifth *FBX* gene distribution type, in which the *FBX* genes from Cluster 1 are over-represented, followed by those from Cluster 4. Similar proportions of *FBX* genes from Cluster 2, Cluster 3, and the orphan gene group are present in this group. The sixth unique group is composed of green algae, which are significantly enriched with the *FBX* genes from Cluster 1 and the orphan gene group followed by those from Cluster 4. Not surprisingly, the red algal outgroup, *Porphyra umbilicalis* (*Pumb*), has a high proportion of *FBX* genes from the orphan gene group followed by genes from Cluster 1, which is similar to green algae.

Because of the variance in the functional constraints of *FBX* genes from the four clusters (Figure 7E), their differential enrichment in six groups of green plants indicates diversifying evolution. To examine this cross-species functional variance, I compared the proportion of *FBX* genes under neutral changes (i.e., *K_a_*/*K_s_* ~ 1) in five groups of embryophytes (the green algae are not compared, due to their high enrichment of orphan genes and low number of duplicated members). Surprisingly, when compared to Brassicales, the other four groups of embryophytes have either similar or significantly lower numbers of neutrally evolving *FBX* genes in Clusters 1–3 (Figure 8B). In Cluster 4, more neutrally evolving *FBX* genes are identified in non-Brassicale rosids than in Brassicales. Although Poacea plants, like Brassicales, contain the second highest number of *FBX* gene family members (Figure 1C and Figure 5A, Appendix A), they, unlike Brassicales, possess the lowest fraction of neutrally evolving *FBX* genes among the five groups of embryophytes compared (Figure 8B), suggesting stronger functional constraints of *FBX* genes in this group than the other flowering plants.

To further examine the functional divergence of the four clusters of *FBX* genes across 111 plants, I clustered the subfamilies and plants based on the number of *FBX* paralogs of a plant in each subfamily. Consistent with the high proportion of neutrally evolving genes in Cluster 1, *FBX* genes barely form large subfamily or plant groups except in the Brassicales and Poaceae species. The absence of connections among large groups of *FBX* subfamilies in the remaining plants indicates their independent origins. The small blocks formed in Brassicales and Poaceae plants may result from as-yet-unidentified WGD events (Figure 8C).

Clustering assays of the remaining three clusters of *FBX* genes are more informative regarding their functional divergence. Consistent with the over-representation of Cluster 2 *FBX* genes in Brassicales (Figure 8A), 163 out of 195 (83.5%) *FBX* subfamilies are enriched in Brassicales but dramatically depleted in other plant groups (Figure 8D). Such a skewed distribution confirmed the selective birth and death pattern of *FBX* genes seen in Cluster 2 (Figure 4B and Figure 6B). Among all four clusters, the species clustering of Cluster 3 best matches the phylogenetic relationship of the six plant groups (Figure 8A,E and Appendix A), suggesting that the expansion of the *FBX* gene family in this cluster likely plays an important role(s) in species adaptation. The subfamilies in Cluster 4 likely represent the core plant *FBX* gene group, which can be subdivided into intermediate- and multiple-copy subgroups (Figure 8F,G). Interestingly, their kernel density curves calculated based on the fraction of plant species containing single copy genes in each subfamily, which was referred as Single-Copy Percentage (SCP) [40], match well the SCP density curves of the intermediate- and multiple-copy core angiosperm families (Figure 8G, [40]). However, in contrast to the skewed SCP distribution of the complete set of core angiosperm families [40], the overall SCP density curve of the Cluster 4 *FBX* subfamilies fits Weibull, normal, and logistic random distribution models with high goodness-of-fit values (Appendix A; Kolmogorov–Smirnov test, *D* < 0.14 and *p* > 0.3 for all three models), further suggesting a random birth-and-death mechanism as seen in their presence/absence (Figure 4D,H) and size distribution (Figure 6D,H) patterns across 111 plant genomes. Similarly, a normal distribution can be modeled to fit the SCP density curve of Cluster 3 *FBX* subfamilies (Appendix A).

### 2.6. Core Arabidopsis FBX Proteins Interact with Arabidopsis Skp1-Like 1

Our previous phylogenetic and functional studies on the *Arabidopsis Skp1-Like* (*ASK*) family argued that *ASK1* is the predominant *ASK* member in Arabidopsis [23,61]. Given the high frequency of Cluster 4 *FBX* genes in plants, I hypothesized that the encoded Arabidopsis FBX proteins within this cluster, designated as the core Arabidopsis FBX (CAF) proteins, bind to ASK1 to assemble an active SCF complex. There are, in total, 129 CAFs present in 93 out of 95 subfamilies in Cluster 4 (Appendix A). Supporting the core activities of CAFs, functionally characterized Arabidopsis *FBX* genes are significantly enriched in this group (Appendix A; [39]). To test the interaction of CAF proteins with ASK1, I cloned the coding sequences of 56 members into a yeast two-hybrid bait vector. Among these, 55 (98.2%) belong to a group of 103 arbitrarily assigned CAF genes in a previous phylogenetic study on 18 plant genomes [15] and 34 out of 56 (60.7%) have been previously reported to physically interact with ASK1 (Figure 9A, Appendix A). Using a yeast two-hybrid assay, 52 and 49 members demonstrated a positive interaction with ASK1 and ASK2 proteins, respectively, in yeast (Figure 9A).

To address whether the remaining untested CAFs likely interact with ASK1, I examined the phylogenetic relationship of 129 CAFs, 102 (79%) of which have been shown to interact with ASK1 either in this work or in previous studies (Figure 9A; see Appendix A for detailed references). Interestingly, except for TIR1/AFB1-3, AFB4/5, and COI1, all the other CAFs are clustered into a single group. Although the CFK1/2-containing subclade branches multiple times to form distant nodes toward SKIP11, many CAFs, including all 27 untested ones, form either a subfamily with two to three members possessing short branches, or are similarly distant with one another (Figure 9B). Therefore, the CAF FBXD sequences are not significantly diversified. Given the strong interaction of TIR1/AFB1-3, AFB4/5, and COI1 with ASK1 (Appendix A), and the well-supported phylogenetic clustering of 27 untested CAFs with the remaining 95 ASK1-interacting CAFs within short branches (Figure 9B), it can reasonably be inferred that ASK1 interaction is a key biochemical feature of CAF proteins.

## 3. Discussion

### 3.1. Diverse Evolutionary Processes of the Plant FBX Genes

Gene duplication has been broadly accepted as the driving force for genome innovation since Ohno’s seminal book published in 1970 [62]. Such mechanisms are particularly robust in plant genome evolution, as evidenced by their high content of gene duplicates [43]. In addition to the presence of diverse gene duplication mechanisms that differ in scale and frequency (WGD vs. SSD), the fate of a gene duplicate also varies dramatically, either disappearing through pseudogenization, or being retained in a genome through numerous random or selective processes, such as genetic/genomic drift, gene dosage advantage, subfunctionalization, dosage balance, paralog interference, neofunctionalization, and escape from adaptive conflict (as reviewed in [43,63]). However, the process by which a new gene duplicate is retained in a genome following its birth (e.g., random vs. selective) is yet to be determined. Given the discovery of both selective and random distributions of subfamilies and the number of *FBX* genes in 111 plant genomes (Figure 4 and Figure 6), it seems that plant *FBX* genes were fixed in genomes via multiple processes.

Through clustering analysis, four clusters of plant *FBX* genes can be clearly distinguished and are seen to possess different retention rates, functional constraints, and phylogenetic distributions. The significantly low retention of WGD duplicates in Cluster 1 *FBX* genes is evidenced by its high proportion of SSD duplicates and weak evolutionary constraints of WGD duplicates (Figure 7). However, given the stronger evolutionary selection of Cluster 1 *FBX* genes in Poaceae species than in other green plants (Figure 8A,B), it seems that they are more evolutionarily beneficial within this group. It was previously reported that the novel daughter copy of rodent duplicates initially experienced accelerated evolution before the rate reduced to prior duplication levels, a process that is independent of the mechanism of duplications [64]. The similar retention rates of WGDs between Poaceae and the remaining plants in Cluster 1 indicate that duplicates, particularly SSDs, are fixed more rapidly (*K_a_*/*K_s_* < 1) in Poaceae than in other plants, where a higher proportion of *FBX* genes are under neutral changes (Figure 8B). It is not yet known whether artificial selection plays any role in this process, because thus far, most Poaceae plants analyzed are domesticated crops. In addition, it has been suggested that large quantities of transposable elements and annual growth habits drive fast evolution of the Triticeae genomes [65]. Consequently, a significantly high proportion of *FBX* pseudogenes were identified in *Taes* (wheat) and *Hvul* (barley) genomes. A similar fast evolution of *FBX* genes could also be a cause of the high rates of *FBX* pseudogenes in *Zmay* and *Msin* (Figure 1C). It will be intriguing to investigate the role of *FBX* genes in improving the agronomic traits of Poaceae crops given the diverse functions of known *FBX* genes [39].

The influence of potential purifying selection or genomic drift on the expansion of *FBX* genes is also evidenced in Cluster 2, in which neither the subfamilies nor the number of *FBX* genes can be found to distribute stochastically in 111 plants (Figure 4B,F and Figure 6B,F). Clustering analysis clearly demonstrated that this group of *FBX* genes is specific to Brassicales. Such a biased enrichment could be ascribed to the specific α- and β-WGDs in this group of plants. Consistent with this notion, K-Pg boundary WGD duplicates are over-represented in Cluster 2 (Figure 7D), and the *FBX* genes from this group are highly enriched in Brassicales (Figure 8A,D).

The diverse evolutionary processes of plant *FBX* genes can be further reflected by the random distribution patterns of members in Clusters 3 and 4, which are evidenced by the stochastic distributions of the number of species per subfamily (Figure 4C,D,G,H), the number of *FBX* genes per species (Figure 6C,D,G,H), and the SCPs of the subfamilies (Figure 8G, Appendix A). Such distributions were not expected, as both clusters enriched more known Arabidopsis *FBX* genes than did Clusters 1 and 2 (Appendix A). The presence of more functionally characterized *FBX* genes may indirectly suggest the presence of high functional constraints that could make them relatively easy to be identified through functional genomic studies. Previous studies suggested that strong selection prevents the SCPs of core angiosperm families from being stochastically distributed among 36 flowering plants [40]. If so, why did these strong functional constraints not skew the SCPs of core *FBX* gene subfamilies from being random? This could be attributed to the specific biochemical function of the *FBX* proteins. The *FBX* protein is composed of an N-terminal FBXD to bind to the Skp1 protein, and a C-terminal substrate-binding region (SBR) that associates with a substrate that is to be ubiquitylated [7,48]. In this work, I biochemically demonstrated that 79% of Arabidopsis FBX proteins from Cluster 4 bind to ASK1 (Figure 9A, Appendix A). Further phylogenetic analysis suggests that all Arabidopsis FBX proteins in this cluster should interact with ASK1 (Figure 9B). Previous work and this study have revealed a co-evolutionary link between the FBXD and the SBR of an FBX protein (Figure 9B, [66]). The unanimous interaction of Cluster 4 Arabidopsis FBX proteins with ASK1 indicates that multiple subfamilies of Cluster 4 may recognize a similar group of substrates, thus reducing their functional constraints and allowing them to be eliminated in plant genomes in a stochastic manner. Indeed, functional and biochemical studies have demonstrated that both TIR1/AFB1-3 and AFB4-5 subfamilies can work as auxin receptors [55,67], while the KMD1/2 and KMD3/4 subfamilies are both involved in the cytokinin signaling pathway by targeting type-B Arabidopsis response regulators (ARRs) for degradation [68]. Further investigation will be required to ascertain whether any other subfamilies, such as the TLP *FBX* proteins (Figure 9B), may also possess similar biochemical function(s).

### 3.2. Purifying and Dosage Balancing Selection Models of the Plant FBX Gene Superfamily

Given the above data, I herein present dual evolutionary models to explain the diverse evolutionary processes controlling plant *FBX* gene abundance by considering that the duplications of most *FBX* subfamilies are detrimental rather than beneficial to plant fitness, and that the active *FBX* genes are subject to substrate dosage balancing selection.

First, given that the orphan *FBX* genes are present at the highest rates in the algal *FBX* families, most, if not all, plant orphan *FBX* genes could arise de novo (Figure 8A and Appendix A [69]). In analogy to nonsynonymous mutations in a protein coding gene that are determined by the mutation rate of a population, I hypothesized that the rate of de novo *FBX* gene synthesis is plant genome-specific, which is consistent with the nonrandom distribution of orphan genes (Appendix A). The strong correlation between the numbers of orphan and Cluster 1 *FBX* genes suggests that members of the latter group could originate from prior de novo synthesized orphan *FBX* genes primarily through SSDs (Figure 5B,C and Appendix A). However, like deleterious nonsynonymous nucleotide mutations within a protein coding gene, purifying selection then prevents the expansion of Cluster 1 *FBX* genes in plant genomes due to their harmful effect. Supporting this notion, over-expressing an *FBX* gene in Arabidopsis is generally more detrimental to plant growth and development than knocking out its expression (unpublished data). Therefore, although Cluster 1 contains 5523 *FBX* subfamilies, many are rare, indicative of strong negative selection against their distribution in plant genomes (Figure 4A and Figure 8C).

However, like the genetic drift of a rare allele of a gene under certain conditions (e.g., small population size), the copy number of these rare *FBX* subfamilies in a plant genome (i.e., small scale) could rapidly increase. For example, if these rare *FBX* genes retain low or no expression, their potential detrimental effect will be eliminated and become neutral. Our previous study demonstrated that the Arabidopsis STSP *FBX* genes remain at low levels of expression due to epigenetic silencing [41]. It could therefore reasonably be deduced that maintaining these rare subfamilies in an epigenetically silenced state would be increasingly difficult if they spread into increasing numbers of plant genomes, due to diverse genome structures and transcriptional regulations. Therefore, although the copy number of Cluster 1 subfamilies could increase dramatically in one or a few plant genomes through a drift mechanism, they remain rare in the plant kingdom (Figure 4A, Figure 5C and Figure 8C). The biased enrichment of Cluster 1 *FBX* genes compared to the lineage-specific angiosperm families further suggests that many of these *FBX* genes likely do not play an adaptive role (Appendix A). Not surprisingly, high rates of Cluster 1 *FBX* genes are under neutral changes (Figure 7E). Due to the random nature of SSDs and neutral changes, as well as their potential deleterious effect(s) if actively expressed, the number of Cluster 1 *FBX* genes in plant genomes follows an asymmetric gamma distribution pattern (Figure 6A,E). In some rare cases, such as Cluster 2 *FBX* subfamilies that primarily arose from α- and β-WGDs, their lineage-specific expansion could also result from WGDs. Further investigation will be needed to determine whether the expression of these *FBX* genes is associated with specific regulatory pathways, e.g., epigenomic suppression as seen in Arabidopsis [41].

Neutral evolutionary theory states that nondeleterious gene duplications fixed through genetic drift could also be useful for the adaptation of organisms to their environment [70,71]. Therefore, we cannot eliminate the possibility that some subfamilies within Clusters 1 and 2 could resume their activity and play adaptive roles for the specific growth and development of plants, such as the reducing rate of neutral changing *FBX* genes in Poaceae species (Figure 8B). However, such frequency would be significantly lower than many angiosperm lineage-specific families, because it requires not only conversion of the potential detrimental effect to adaptive function, but also expression activation of the *FBX* duplicates. 

Although the majority of *FBX* gene subfamilies in Clusters 1 and 2 are potentially deleterious, those in Clusters 3 and 4 likely represent beneficial members that are active and under positive or dosage balancing selections. If one subfamily plays an important role in plant fitness, like TIR1/AFB1-3, the frequency with which the subfamily is present tends to increase in plant genomes through positive selection, a process similar to the fixation of beneficial gene alleles in a population. If a daughter subfamily shares a similar function with its parental or other subfamilies, like AFB4-5 with TIR1/AFB1-3, its frequency in plant genomes may also be subject to dosage balancing selections. Like many other gene duplications, new duplicates in these two clusters could also lead to subfunctionalization, neofunctionalization, paralog interference, etc. [43]. However, the number of active subfamilies and *FBX* genes from these two clusters in each plant genome would remain subject to dosage balancing selection, which is primarily determined by the pool of SCF substrates. The large variation of these substrates at both expression and functional levels may thus ultimately give rise to the stochastic distribution of these *FBX* genes in plant genomes (Figure 4C,D and Figure 6C,D).

## 4. Materials and Methods

### 4.1. CTT Annotation of FBX Genes in 111 Plant Genomes

The CTT program was used to automatically find a nearly complete set of *FBX* genes in each plant genome by predicting the presence of an FBXD in a previously annotated protein sequence and re-annotating new *FBX* loci in the genome [50]. For each of the 111 plant genomes, the genome sequence dataset, the general feature format 3 (GFF3) file, and the prior-annotated protein sequence dataset were downloaded from the Phytozome V13 database (https://phytozome-next.jgi.doe.gov; Appendix A). In order to discover most, if not all, plant *FBX* genes, a customized FBXD HMM profile, named AO_FBX.hmm, was constructed based on a reference FBXD sequence dataset (see below) of the Arabidopsis and rice FBX proteins retrieved from our previous work [15]. Additionally, the HMM profiles of five types of FBXD sequences including F-box, F-box-like, F-box-like_2, F-box_4, and F-box_5 from Pfam 32 (https://pfam.xfam.org) were also used.

CTT predicted the presence of an FBXD in a prior-annotated protein sequence by HMMER, which found a fragment in the protein sequence that could be aligned with one of the six FBXD HMM profiles with an e-value ≤ 1. The predicted FBXD sequences from 111 plants were combined with the seed sequences of five Pfam 32 FBXD HMM profiles and used to search new *FBX* loci in 111 genomes based on a Closing-Target-Trimming algorithm wrapped in the CTT program [50]. Briefly, the combined nonredundant FBXD sequences were used as queries to tBLASTn each genome sequence dataset. A 10 kilobase pair region of genomic sequence flanking each tBLASTn hit, which is not overlapped with prior-annotated *FBX* loci, was used for gene model prediction by GENEWISE [51]. If the predicted protein-coding gene overlaps with a previously annotated *FBX* locus, the corresponding genomic DNA sequence was trimmed. Following six iterations, only if a new gene possessed a GENEWISE score >50 was it further searched by HMMER to find the presence of an FBXD in its encoded peptide sequence. If premature stop codon or frame shift mutations were discovered by GENEWISE, the corresponding locus was considered an *FBX* pseudogene. 

### 4.2. Construction of AO_FBX.hmm

In total, 1403 FBXD sequences from Arabidopsis and rice FBX proteins were retrieved from our previous work [15]. FBXD sequences shorter than 36 amino acids (80% of the mean length) were removed to reduce the likelihood of predicting truncated FBXD fragments. CD-HIT was used to further filter out redundant sequences [72]. The resulting 1341 nonredundant FBXD sequences were aligned by MAFFT [73], converted to sto alignment format, then used to construct the AO_FBX.hmm profile using hmmbuild from the HMMER3 package (http://hmmer.org).

### 4.3. Orthologous Group Analysis

To identify *FBX* subfamilies, the complete set of FBX proteins was subjected to OrthoMCL analysis to identify subgroups that find each other reciprocally based on sequence similarities [54]. An inflation value of 1.5 was applied to yield a total of 5858 OrthoMCL groups, where most known *FBX* subfamilies were identified to be in the same group. The remaining *FBX* genes that are not assigned to any OrthoMCL groups were considered as orphan genes in each plant genome.

### 4.4. Multi-Dimensional Clustering Analysis

Previous studies divided gene families or subfamilies based simply on the number of species present in an OrthoMCL group [15,40]. To better interpret the relationship among *FBX* subfamilies supported with strong statistical evidence, a multi-dimensional clustering approach was developed. A data matrix was constructed with five columns containing the values of the number of species (cnt), sum of between-species phylogenetic distance (dist), number of missing species between the two most distant species (gap), and the maximum and mean number of paralogs in each subfamily. The resulting dataset was subject to a k-means clustering analysis. The R package “ConsensusClusterPlus” was applied to better determine the cluster number and clustering confidence, using the following settings: maxK = 9, reps = 1000, pItem = 0.8, pFeature = 1, innerLinkage = “average,” finalLinkage = “average,” clusterAlg = “km,” distance = “Euclidean.” Based on the relative change in area under the cumulative distribution function (CDF) curve, a number of four clusters was determined to be optimal [59,74].

### 4.5. Determining Phylogenetic Distance of Species in an FBX Subfamily

A complete bifurcate species tree was constructed based on the phylogenetic tree available at Phytozome V13 (https://phytozome-next.jgi.doe.gov). The newick tree file was converted into a 111 × 111 matrix-representation (mr) matrix using the “compute.mr” function in the R package “phytools” [75]. A subset of the mr matrix for each *FBX* subfamily was built by retrieving the rows and columns from the complete mr matrix that contains the names of each species in the subfamily. Hence, the phylogenetic distance of species in the subfamily was represented by the sum of distance of the matrix.

### 4.6. Calculating the Number of Missing Species in an FBX Subfamily

For each subfamily, the species names were retrieved and aligned with the row names of the 111 × 111 mr matrix. The two species that separated the furthest were considered the two distant ends of the subfamily. The number of species missing in between was calculated as the number of missing species of the *FBX* subfamily.

### 4.7. Statistical Modeling of Data Distribution

The kernel density curve and the histogram of data distribution for the number of species per subfamily or the number of *FBX* members per species in each of the four clusters were calculated using the “geom_density(alpha = 0.2, fill =“#FF6666”)” and “geom_histogram(aes(y =..density..))” functions, respectively, in the R package “ggplot2”. To model the kernel density curve, the “fitdist” function from the R package “fitdistrplus” was used to test the “goodness-of-fit statistics” for different statistical models. The best fitted model was further selected to test its goodness-of-fit to the observed unique value using the Kolmogorov–Smirnov test by “disc_ks_test” in the R package “KSgeneral”.

The retention rates of WGD duplicates in the four clusters of *FBX* genes or the complete set of paralogs in 26 flowering plants were plotted by a function of mean *K_s_* values, which were used to estimate the age of three different WGDs. Each set of data points was fitted to a power-law function using the “nls*”* (nonlinear least squares) function as described in [40]. The *x*^2^ value was calculated to examine the goodness-of-fit for the fitted model.

To examine the stochastic distributions of subfamily SCPs in both Clusters 3 and 4, each kernel density was tested and verified using “fitdist” and “disc_ks_test” as described above.

### 4.8. Estimation of K_s_-Based Duplication Time in 27 Flowering Plants

To estimate *K_s_*-based duplication times, each *FBX* gene in a subfamily was selected for comparison with all its paralogs to calculate pairwise *K_s_* values using the codeml program available in the PAML4 package [76]. For each pair of homologous genes, the coding sequences were aligned based on their protein sequence alignment obtained via T-Coffee [77] and used as an input file to run a pairwise model of codeml [76] to obtain the *K_s_* value. Only the paralogous pair that yielded the lowest *K_s_* value was considered as the timing of the duplication event [40]. 

The resulting *K_s_* values were used to estimate the duplication events of *FBX* genes in 27 flowering plants that were also studied in Li et al. [40]. A specific duplication event was assigned to an *FBX* gene if its *K_s_* value fell within the low and high boundary of *K_s_* values that had been estimated to define each of the four duplication events, including SSD, recent, KT, and ancient WGDs, in the 27 plant genomes through Gaussian mixture modeling by Li et al. [40]. The mean *K_s_* values of *FBX* genes assigned to each of the three WGD events from each plant genome were used to model the dynamic retention processes of WGDs.

### 4.9. Paralogous Neutral Evolution Test

All paralogous pairs in each subfamily were used to test their neutral evolutionary processes if the resulting *K_s_* value was less than 5 and greater than 0. The nucleotide sequence alignment back-translated from the protein sequence alignment of each pair was obtained as above via T-Coffee [77]. However, the aligned nucleotide sequences were subject to codeml analysis twice, with the *K_a_*/*K_s_* ratio either fixed at 1 or free [76]. The maximum likelihood (ML) values ML1 and ML2 from the two runs were extracted to obtain a likelihood ratio based on the formula, LR = 2 (lnML1–lnML2). If LR was less than 2.71 (5% significance for *x^2^* distribution with one degree of freedom) [76], the *K_a_*/*K_s_* value was determined to be close to 1, which is a feature of protein coding genes that are under neutral evolutionary processes.

### 4.10. Plasmid Construction and Yeast Two-Hybrid Analysis

The coding sequences of 56 *F-box* genes and *ASK1/2* were PCR amplified either directly from Arabidopsis seedling cDNAs or from a cDNA clone obtained from the Arabidopsis Biological Resource Center (https://abrc.osu.edu). Each *FBX* CDS fragment was fused in-frame to the GAL4-binding domain (BD) coding sequence in the yeast two-hybrid bait vector pGBK-T7 to produce BD-FBX fusions in yeast. The CDS fragment of *ASK1/2* were cloned into the *Eco*RI-*Bam*HI sites of the yeast two-hybrid prey vector pGAD-T7 to make activation domain (AD)-ASK1/2 fusions. Each *FBX* bait vector was transformed into the haploid yeast strain AH109, which was then mated with another haploid yeast strain, Y187, pre-transformed with one of the three prey vectors pGAD-T7, pGAD-T7-ASK1, or pGAD-T7-ASK2 to generate a diploid yeast strain co-expressing the bait and prey proteins.

Following mating, two independent diploid yeast clones were streaked and grown on quadruple synthetic dropout medium (SD-Leu-Trp-Ade-His, Takara Bio, Mountain View, CA, USA) containing 20 μg/mL X-α-gal (Takara Bio) to test the interaction of each *FBX* protein with ASK1/2 in yeast. The growth of yeast cells co-expressing each BD-FBX protein with AD alone was used as a negative control to test for GAL4 activation by the *FBX* protein alone. Yeast cells with blue coloration indicate a strong interaction between the two proteins tested.

### 4.11. Sequence Alignment and Phylogenetic Analysis

The CTT-predicted FBXD sequence of each Arabidopsis *FBX* protein in Cluster 4 was retrieved and combined for sequence alignment analysis using MAFFT [73] and MUSCLE [78]. A consensus alignment was resolved using Trimal (-conthreshold 0.5) [79]. The resulting sequence alignment was converted to a nexus format for Bayesian inference analysis to assay the phylogenetic relationship of *FBX* proteins using MrBayes 3.2.7a (rates = gamma, covarion = no, aamodelpr = mixed, four Markov-chain Monte Carlo strands, 10^7^ generations, average standard deviation of split frequencies = 0.01) [80]. A consensus phylogenetic tree was constructed after excluding an initial burn-in of 25% of the samples.

## Figures and Tables

**Figure 1 ijms-22-00871-f001:**
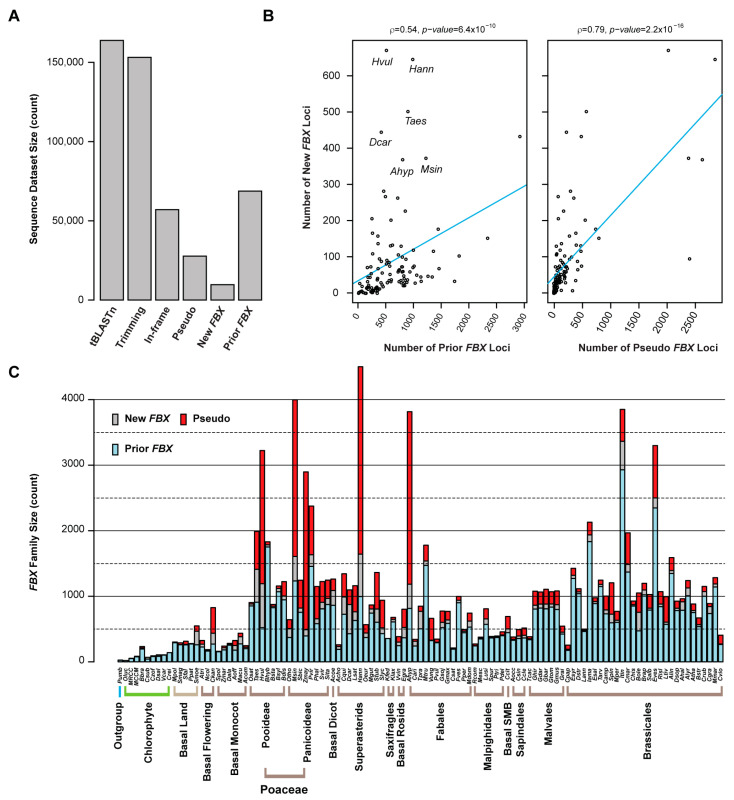
Size variation of the *FBX* superfamily across 110 green plants and one red alga. (**A**) Step down number changes of sequences identified in similarity-based Closing-Target-Trimming (CTT) annotations of new *FBX* loci in 111 plant genomes. (**B**) The number of new *FBX* loci is correlated with the number of previously annotated *FBX* genes and pseudo-*FBX* loci identified by GENEWISE in each genome. Five outliers are indicated in the left panel. A correlation assay was performed using Spearman’s rank-order test. (**C**) Size comparison of the *FBX* gene superfamily across different plant genomes and phylogenetic levels. The number of prior, new, and pseudo-*FBX* genes are separately indicated in each genome. The abbreviations for species name are as defined in Appendix A.

**Figure 2 ijms-22-00871-f002:**
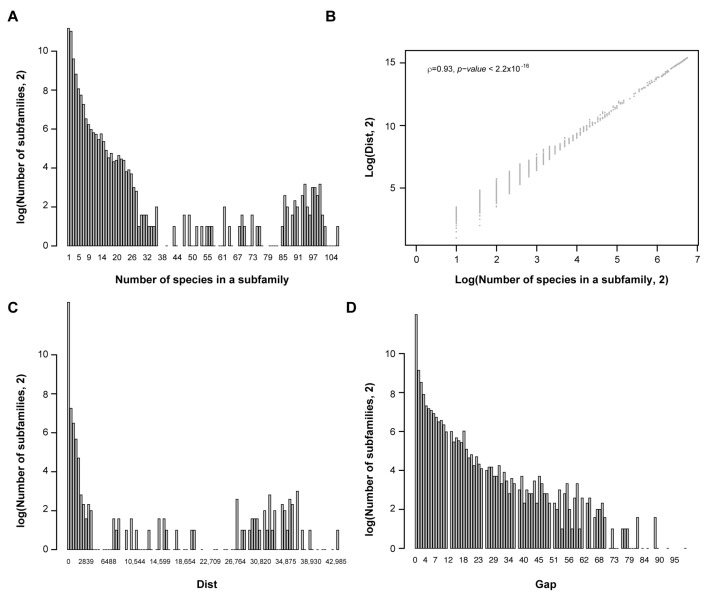
Multi-dimensional comparison of *FBX* subfamilies. (**A**) Bimodal distribution of *FBX* subfamilies in accordance with the number of plants in each subfamily. (**B**) Phylogenetic distance better separated the lineage-specific and core *FBX* gene groups. (**C**) Phylogenetic distance is better correlated with the number of species in conserved subfamilies. (**D**) The number of subfamilies declines exponentially in accordance with the number of missing species in each subfamily.

**Figure 3 ijms-22-00871-f003:**
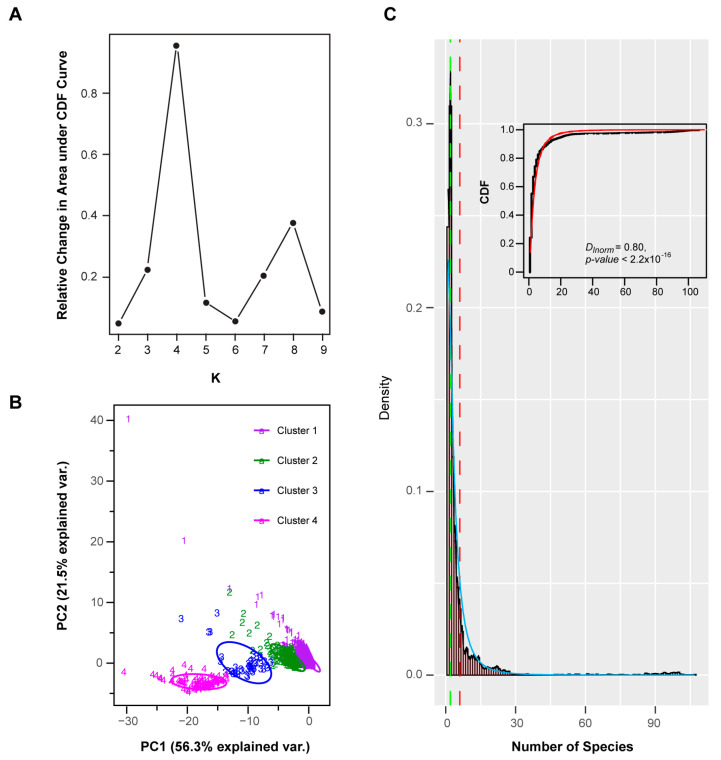
Multi-dimensional features of the *FBX* subfamilies. (**A**) A delta area plot analyzed by ConsensusClusterPlus based on the number of species, phylogenetic distance, missing species, and the maximum and mean number of *FBX* genes per plant of each subfamily demonstrates four optimal k-mean clusters of *FBX* subfamilies. (**B**) A bi-dimensional plot showing the first two dimensions of a principal component analysis (PCA) of multi-dimensional features of the *FBX* subfamilies. PC1 accounts for 56.3% of the variance between individuals and PC2 accounts for 21.5%. Colored data points indicate the four clusters obtained from the analysis in (**A**). (**C**) The distribution of *FBX* subfamilies is biased toward lineage-specificity more strongly than expected under a random model, suggestive of strong selection. Dashed red and green lines indicate the mean and mode number of species in a subfamily, respectively. Solid black and cyan lines indicate empirical and expected density curves under a log-normal distribution, respectively. The inset shows the curves of the empirical (black dots) and expected (red line) cumulative distribution function (CDF) under a log-normal distribution model. The Kolmogorov–Smirnov test indicates that the empirical and expected CDF curves do not fit.

**Figure 4 ijms-22-00871-f004:**
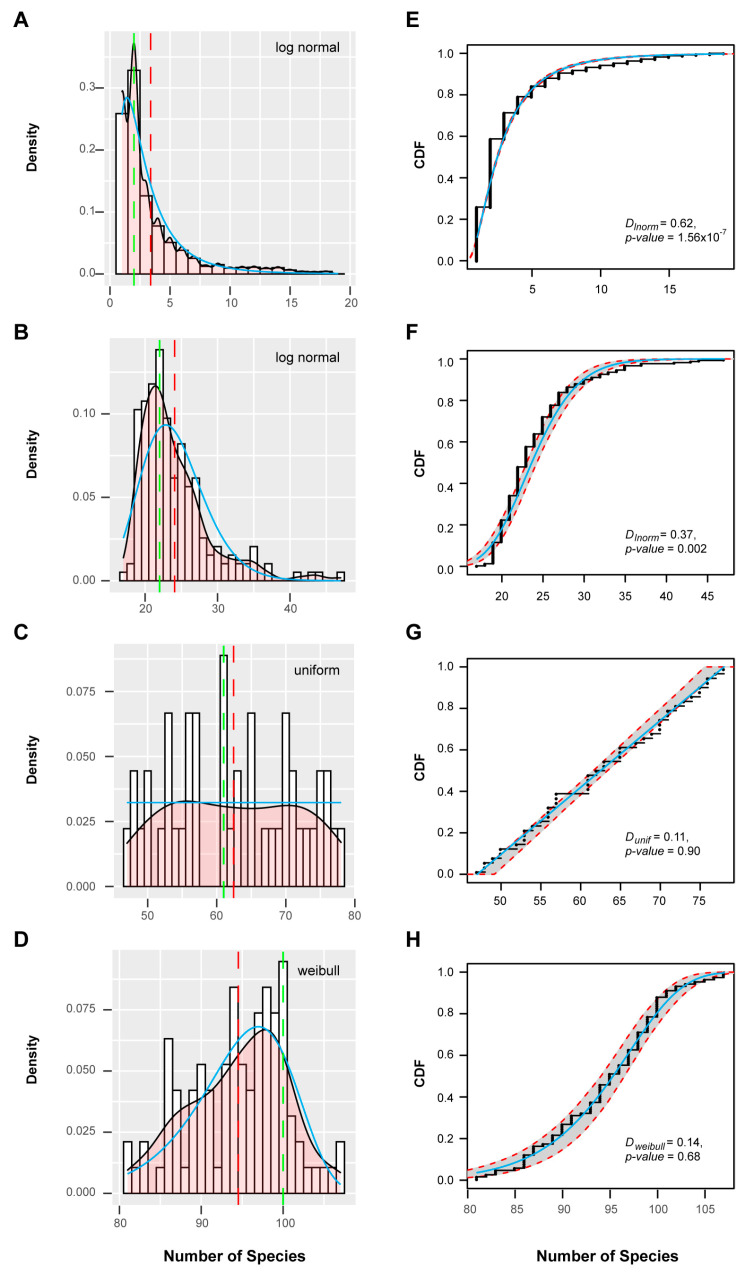
Statistical modeling of subfamily distributions in accordance with the number of species in four clusters of *FBX* genes. (**A**–**D**) Density distributions of subfamilies with a different number of species in Clusters 1 (**A**), 2 (**B**), 3 (**C**), and 4 (**D**). Black and cyan lines represent the empirical and expected data, respectively. The statistical model indicated in each panel was the best fitting model calculated using the “fitdistrplus” R package. Dashed red and green lines indicate the mean and mode number of species per subfamily, respectively. (**E**–**H**) Curves of empirical (black dots) and expected (cyan line) CDF for the number of species per subfamily in Clusters 1 (**E**), 2 (**F**), 3 (**G**), and 4 (**H**). Red dashed lines and shaded bands mark a 95% confidence interval. The Kolmogorov–Smirnov test result is included in each panel to show the goodness-of-fit of the statistical model.

**Figure 5 ijms-22-00871-f005:**
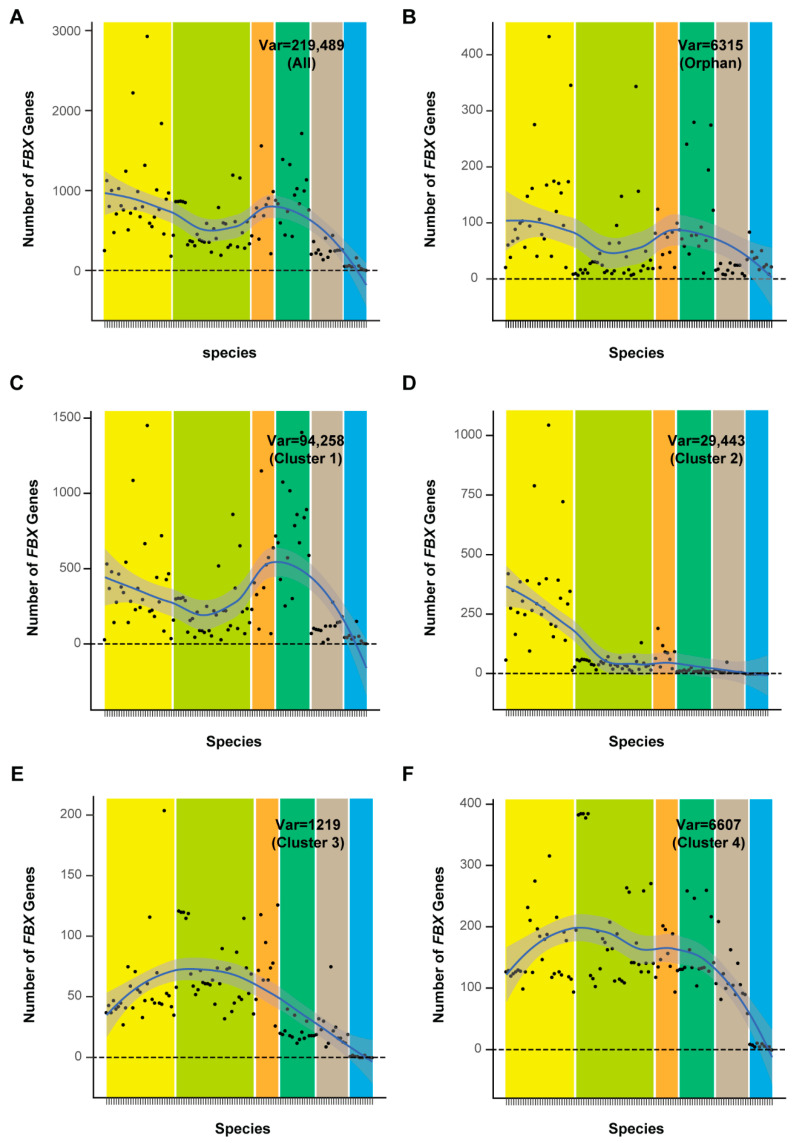
Variation in the number of *FBX* genes in different groups in 111 plant genomes. The solid blue line and shaded area represent the LOESS (Local Regression) fits ±95% confidence interval in each panel. The value of number variation is also indicated. The yellow, light green, orange, dark green, gray, and cyan bars ordered from left to right in each panel indicate six groups of plants that belong to Brassicales, non-Brassicale rosids, non-rosid dicots, Poaceae, basal embryophytes, and algae, respectively. The vertical lines in the *x*-axis represent 111 plant genomes that are grouped and listed in an order as in Figure 8A. (**A**) The complete set of *FBX* subfamilies. (**B**) The group of orphan *FBX* genes. (**C**) Cluster 1 *FBX* subfamilies. (**D**) Cluster 2 *FBX* subfamilies. (**E**) Cluster 3 *FBX* subfamilies. (**F**) Cluster 4 *FBX* subfamilies.

**Figure 6 ijms-22-00871-f006:**
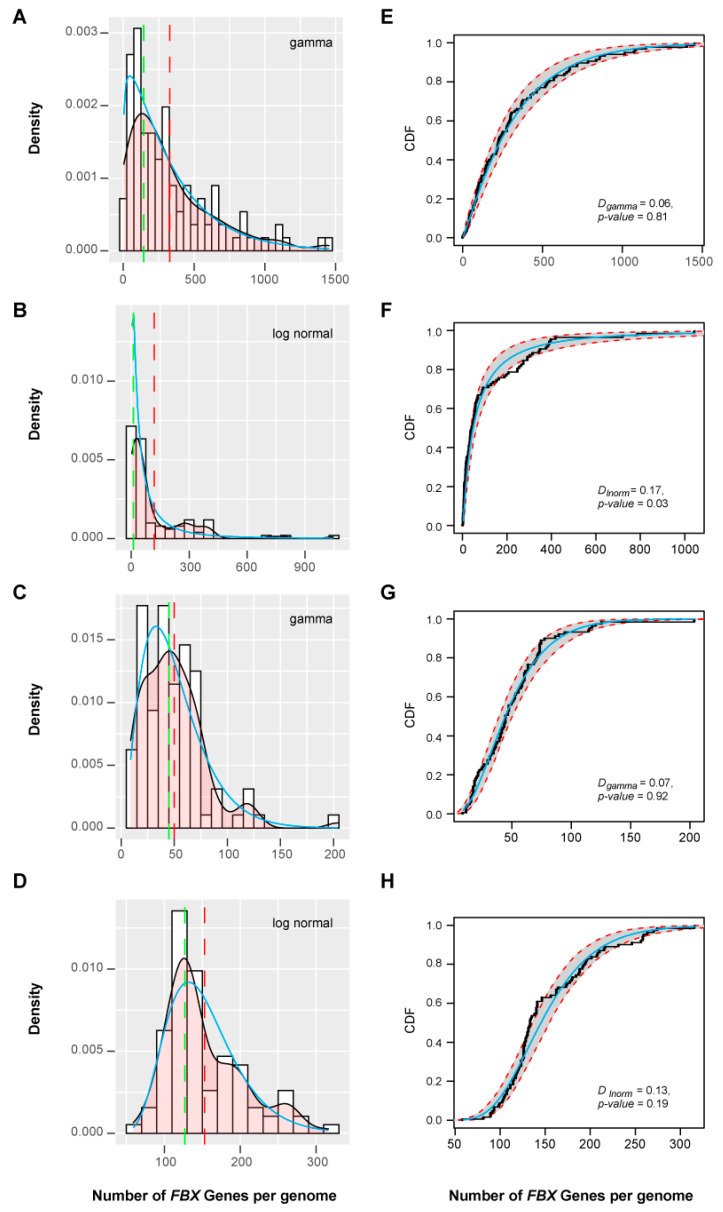
Statistical modeling of the variation in *FBX* gene numbers per plant genome in four clusters of *FBX* genes. (**A**–**D**) Density curves of number of *FBX* genes per plant genome in Clusters 1 (**A**), 2 (**B**), 3 (**C**), and 4 (**D**). Black and cyan lines represent the empirical and expected data, respectively. The statistical model indicated in each panel was the best fitting model calculated using the “fitdistrplus” R package. Dashed red and green lines indicate the mean and mode number of *FBX* genes per genome, respectively. (**E**–**H**) Curves of empirical (black dots) and expected (cyan line) CDF for the number of *FBX* genes per plant genome in Clusters 1 (**E**), 2 (**F**), 3 (**G**), and 4 (**H**). Red dashed lines and shaded bands mark a 95% confidence interval. The Kolmogorov–Smirnov test result is included in each panel to show the goodness-of-fit of the statistical model.

**Figure 7 ijms-22-00871-f007:**
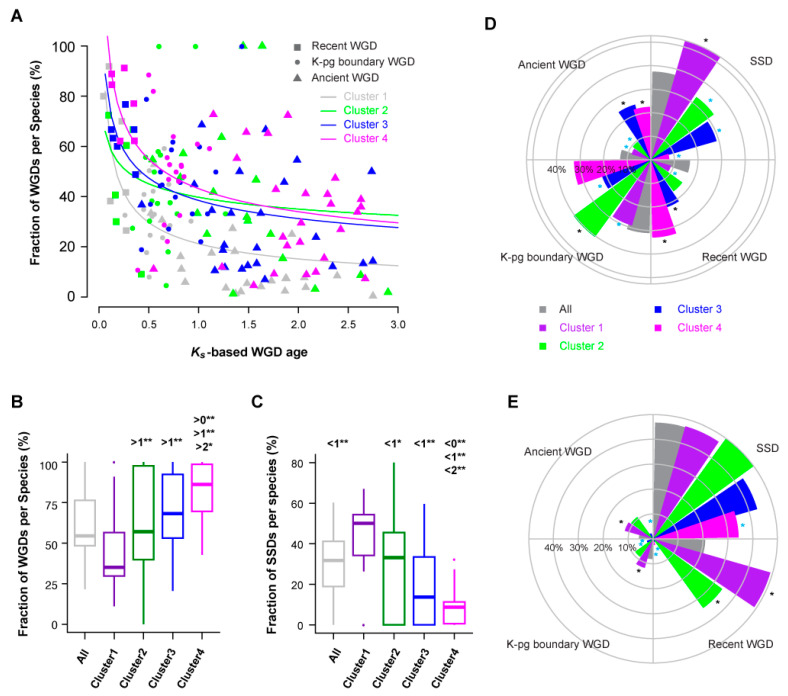
Duplication retention and functional constraint analyses in 27 flowering plants. (**A**) The contribution of whole genome duplications (WGDs) in the total number of duplicated *FBX* genes of each plant genome can be modeled as a power-law function of *K_s_*-based age. The detailed *x*^2^ goodness-of-fit data for each group of *FBX* genes can be found in Appendix A. (**B**,**C**) Multiple statistical comparisons showing unequal contributions of WGDs *(B)* and small-scale duplications (SSDs) *(C)* in expanding the size of different groups of *FBX* genes. Single and double asterisks indicate *p*-value < 0.05 and < 0.01, respectively. *P*-values were calculated based on the Kruskal–Wallis rank sum test followed by Dunn’s test with Benjamini–Hochberg multiple testing correction using the R package “dunn.test.” The numbers 0, 1, and 2 indicate the groups of total, Cluster 1, and Cluster 2 FBX subfamilies, respectively. (**D**) Contribution of differential duplication events to the expansion of *FBX* genes in five indicated groups. Duplication events were inferred based on *K_s_*-based duplication ages in different plant genomes as in (**A**) and divided into “Ancient,” Cretaceous-Paleogene boundary (“K-Pg”), and “recent” WGDs, and SSD duplications. “All” indicates the total *FBX* subfamilies. Black and cyan asterisks designate the statistically significant over- and under-representation, respectively, of duplicates of a specific group derived from a denoted duplication event, compared with those of the full set from the same duplication event, as calculated by Fisher’s exact test with Bonferroni multiple-testing corrections. (**E**) Neutral evolution comparison of different groups of *FBX* genes under four different types of duplication events. The five groups of *FBX* genes are color coded as in (**D**). Black and cyan asterisks are also used to designate statistically significant comparison as in (**D**).

**Figure 8 ijms-22-00871-f008:**
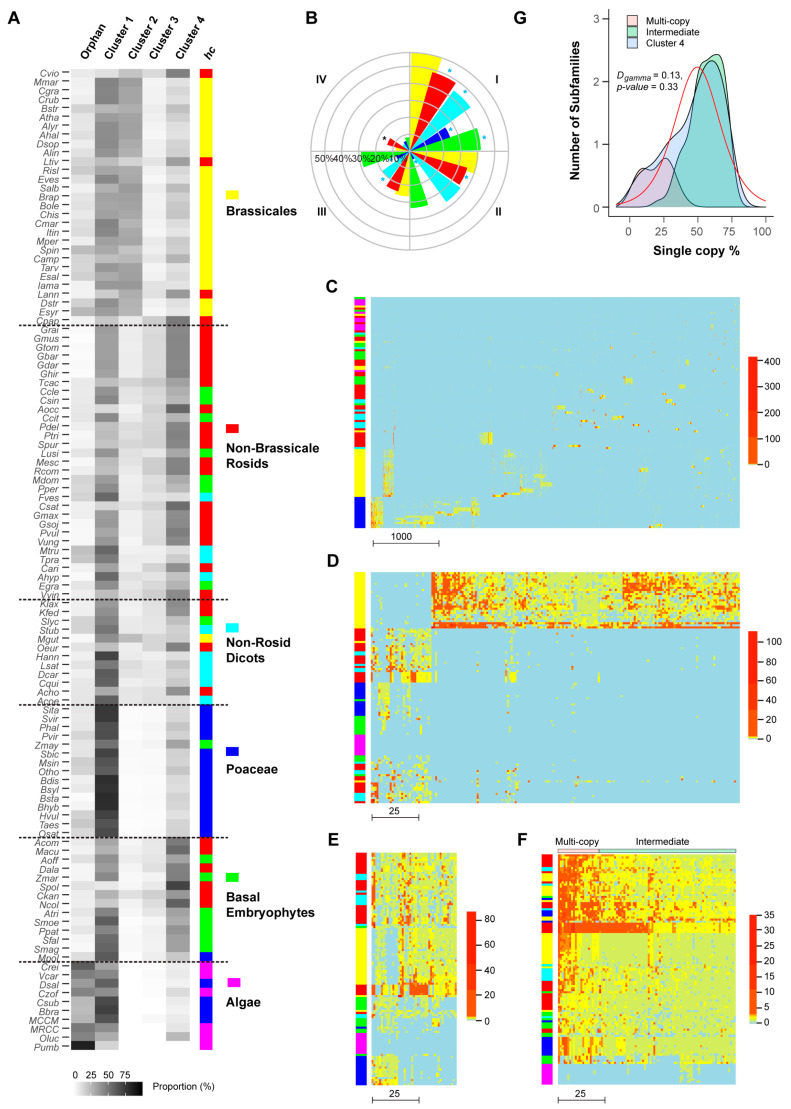
Landscape of *FBX* distributions at different phylogenetic and conservation levels. (**A**) The contributions of five independent groups of *FBX* genes in each genome are diversified in a phylogenetic group-dependent manner. The 111 plant species are divided into six phylogenetic groups as indicated. The color coded “*hc*” side bar denotes six hierarchical clusters of the data matrix. The dominant color displayed in each phylogenetic group was selected to indicate the corresponding group. (**B**) A neutral evolution comparison of *FBX* genes from five different phylogenetic groups of embryophytes in four clusters of subfamilies. The groups are color coded as in (**A**). (**C**–**F**) Clustering analysis shows contrasting landscapes of *FBX* distributions in four clusters of *FBX* subfamilies: Cluster 1 (**C**), Cluster 2 (**D**), Cluster 3 (**E**), and Cluster 4 (**F**). Rows represent 111 plant genomes and columns represent the different number of subfamilies that are scaled with the bottom scale bar in each heat map. The bars on the left indicate the six phylogenetic groups of plants color coded as in (**A**). The scale bars on the right show the number of *FBX* genes per genome in each subfamily. (**G**) Single-copy percentage (SCP) distribution of three groups of subfamilies in Cluster 4. The red line shows the expected logistic distribution of SCPs of the entire cluster of subfamilies. The Kolmogorov–Smirnov test result is included to show the goodness-of-fit of the statistical model.

**Figure 9 ijms-22-00871-f009:**
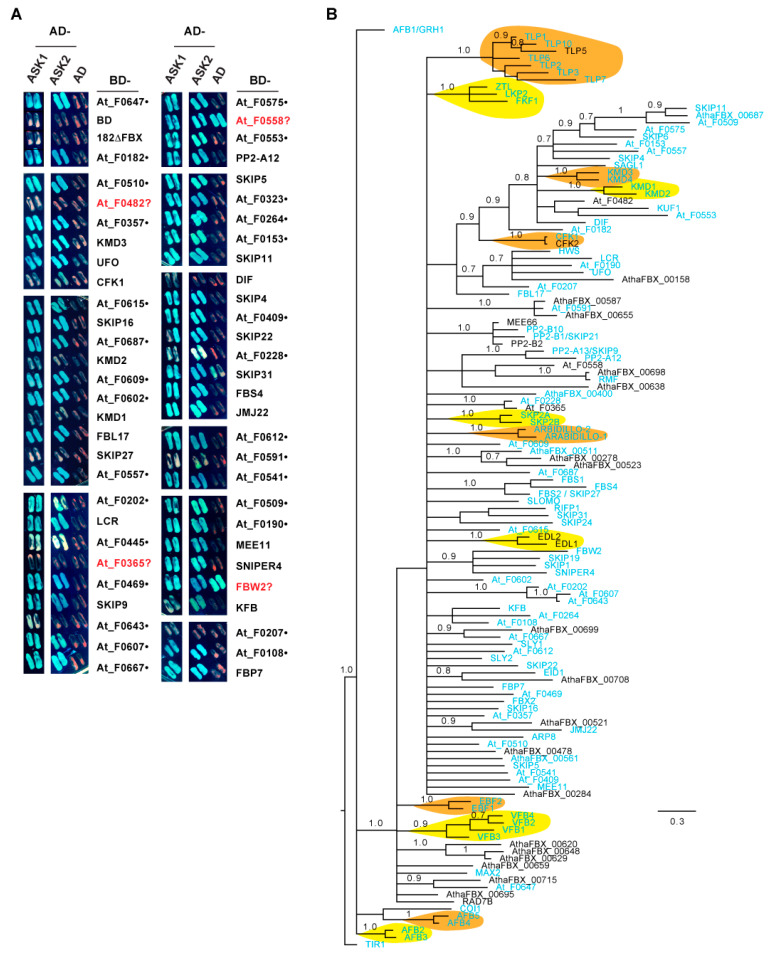
CAF proteins interact with ASK1 and ASK2. (**A**) Yeast two-hybrid analysis of 56 CAF proteins demonstrating their interactions with ASK1 and ASK2. The coding sequence of each gene was PCR amplified and subsequently ligated in-frame into pGBK-T7 (bait) or pGAD-T7 (prey) vectors, which were transformed into yeast strains AH109 and Y187, respectively. Indicated pairs were mated together to produce a diploid yeast fusion. Shown are mated yeast cells grown on a synthetic dropout medium lacking Leu, Trp, His, and Ala (SD-L-W-H-A) and supplemented with 20 μg/mL X-α-Gal. Yeast cells with blue coloration indicate a strong interaction between the two proteins tested. The given name of *FBX* genes with known functions is indicated; uncharacterized genes are labeled with the yeast clone name, which is the *FBX* ID used in a previous study [15]. The accession number of each *FBX* gene can be found in Appendix A. A truncated F-box protein (182ΔFBX), in which the FBX domain (FBXD) has been deleted, and two empty vectors, pGBK-T7 and pGAD-T7, were used as negative controls. The question marks indicate weak or inconclusive interactions. The black dots indicate 29 newly discovered FBX proteins that are able to assemble into an SCF complex through interaction with ASK1/2. (**B**) Phylogenetic analysis demonstrating the monophyletic relationship of all CAF proteins. The tree was constructed based on the FBXD sequences using MrBayes 3.2.7a, running 10^7^ generations until the average standard deviation of split frequencies reached 0.01. The initial 25% of the samples were excluded from resolving the final consensus tree. Posterior probabilities ≥0.7 are shown above each corresponding branch. The FBX proteins with known biochemical interactions with ASK1 (see details in Appendix A) are highlighted in light blue. Known FBX protein subfamilies are shaded in light brown and yellow. The tree was rooted to TIR1 and displayed using FigTree (http://tree.bio.ed.ac.uk/software/figtree/).

## Data Availability

The datasets presented in this study can be found in the article or in the Appendix A. Processed data and R scripts are made available as a GitHub repository at https://github.com/hua-lab/IJMS2021_FBX_Evolution_in_111_Plants.

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
