# Peer review of "Diverse Evolution in 111 Plant Genomes Reveals Purifying and Dosage Balancing Selection Models for F-Box Genes"

_ijms, 2021, doi:10.3390/ijms22020871_

Round 1
Reviewer 1 Report
This is a very well written, clear analysis of F-box protein genes in plant species. This work builds upon previous work by the author and author’s laboratory. It does represent significant advance in analysis of the evolutionary relationship and the forces that balance gene number with functionalization.
I only have a few minor suggestions.
Minor
- Line 58, switch to singular, cell cycle.
- Page 19. I am confused about the analysis of Fbox protein interaction with Skp in Y2H assays. The author writes, “I cloned the coding sequences of 56 members into a yeast two-hybrid… 55 (98.2%) belong to a group of 103 arbitrarily assigned CAF genes in a previous phylogenetic study… and 34 (60.7%) have been previously reported to physically interact with ASK1..”
I don’t understand the percentages given, 55 is 98.2% of 56 proteins test, but the number 34, the 60.7% is the percentage of what? Please clarify.
- Maybe Fig 9B needs an outgroup? A mammalian F box?
Author Response
Response to Reviewer 1’s Comments
Comment 1. This is a very well written, clear analysis of F-box protein genes in plant species. This work builds upon previous work by the author and author’s laboratory. It does represent significant advance in analysis of the evolutionary relationship and the forces that balance gene number with functionalization.
I only have a few minor suggestions.
Minor
Line 58, switch to singular, cell cycle.
Response 1. Thank you for this editorial comment. The word has been revised accordingly.
Comment 2. Page 19. I am confused about the analysis of Fbox protein interaction with Skp in Y2H assays. The author writes, “I cloned the coding sequences of 56 members into a yeast two-hybrid… 55 (98.2%) belong to a group of 103 arbitrarily assigned CAF genes in a previous phylogenetic study… and 34 (60.7%) have been previously reported to physically interact with ASK1..”
I don’t understand the percentages given, 55 is 98.2% of 56 proteins test, but the number 34, the 60.7% is the percentage of what? Please clarify.
Response 2. Thanks for pointing out this problematic sentence. 60.7% is calculated based on the same denominator, which is 56. To make it clear, I have revised the sentence that now reads, “I cloned the coding sequences of 56 members into a yeast two-hybrid… 55 (98.2%) belong to a group of 103 arbitrarily assigned CAF genes in a previous phylogenetic study… and 34 out of 56 (60.7%) have been previously reported to physically interact with ASK1..”
Comment 3. Maybe Fig 9B needs an outgroup? A mammalian F box?
Response 3. Thanks for this comment. The phylogenetic tree shown in Fig 9B does have an outgroup. It is rooted to TIR1, a well characterized Arabidopsis FBX protein that functions as the auxin receptor and it physically associates with ASK1. I believe TIR1 can serve as a good outgroup FBX protein to suggest that most, if not all, FBX proteins in the group bind to ASK1. To clearly stated this outgroup FBX protein, I have added one sentence in the legend of Figure 9B. The sentence states, “The tree was rooted to TIR1 and displayed using FigTree (http://tree.bio.ed.ac.uk/software/figtree/).” (Line 544)
Reviewer 2 Report
The manuscript “Diverse evolution in 11 plant genome reveals purifying and dosage balancing selection models for F-box genes” shows the specific function of F-box genes across a wide range of 111 plant species, based on genomic and phylogenetic comparison, in order to shed new light in gene duplication. The authors focus on the role of FBoX genes in Poaceae family, supporting the role of genome size in gene copy number variation. In addition, a huge statistical analysis was carried out in Cluster 1, Cluster 2, Cluster 3 and Cluster 4 as well as the role of FBoX genes in many species, especially in Arabidopsis. Nevertheless, your Review needs to be improved for publication. Few questions have to be addressed:
Line 52: It would be very useful expanding the subject with the role of UPS in case of abiotic stress in plants. For instance, this reference “The role of ubiquitin and the 26S proteosome in plant abiotic stress signalling” might be cited in order to improvement your sentence, https://doi.org/10.3389/fpls.2014.00135.
Line 100: Please, could you improve your sentences by adding these two manuscripts?
- Whole genome duplications in Plants: Overview from Arabidopsis, doi: 10.1093/jxb/erv432.
- Gene duplication and evolution in recurring polyploidization – diploidization cycle in plants, doi: 10.1186/s13059-019-1650-2.
Line 111: What kind of epigenetic mechanism is involved in transcriptional silencing?
Line 147: I did not find the table S1.
Line 170: I will highlight that the correlation between FBoX genes abundance and genome size is probably derived by the polyploidy levels that occur in a plethora of species. For instance, the Poaceae family are generally polyploid species, and this could explain the abundant of FBoX loci in some species belong to this family.
Line 185: Please, could you indicate which statistical method was used for data analysis?
Line 233: Please, indicate the statistical test used for the analysis.
Line 269: based on the results, It emerges that only the Cluster 1 and cluster 2 presents a cnt value statistically significant compared with the other clusters. It sould be useful to add more details to support your outcomes.
Line 400: Please, could you add more details? For instance, the Poaceae families probably have much more genes in cluster 1 due to gene copy number variation induced by gene duplication?
Line 477: It is very interesting the CAF activity that you suggested. If it were possible, could you add more details? For instance, its binds to ASK1 in promotor region or in gene body?
Line 480: I did not find the Figure 14.
Line 489: I did not find the Table 2.
458: I did not find Figure 12.
Line 478: I did not find the Figure 13.
Line 810: I did not find Figure 10.
Author Response
Response to Reviewer 2’s Comments
Comment 1. The manuscript “Diverse evolution in 11 plant genome reveals purifying and dosage balancing selection models for F-box genes” shows the specific function of F-box genes across a wide range of 111 plant species, based on genomic and phylogenetic comparison, in order to shed new light in gene duplication. The authors focus on the role of FBoX genes in Poaceae family, supporting the role of genome size in gene copy number variation. In addition, a huge statistical analysis was carried out in Cluster 1, Cluster 2, Cluster 3 and Cluster 4 as well as the role of FBoX genes in many species, especially in Arabidopsis. Nevertheless, your Review needs to be improved for publication. Few questions have to be addressed:
Line 52: It would be very useful expanding the subject with the role of UPS in case of abiotic stress in plants. For instance, this reference “The role of ubiquitin and the 26S proteosome in plant abiotic stress signalling” might be cited in order to improvement your sentence, https://doi.org/10.3389/fpls.2014.00135.
Response 1. I appreciate this reviewer’s recognition about the role of the UPS in plant abiotic stress signaling. The recommended reference was cited in the original submission. Please see Reference 12.
Comment 2. Line 100: Please, could you improve your sentences by adding these two manuscripts?
Whole genome duplications in Plants: Overview from Arabidopsis, doi: 10.1093/jxb/erv432.
Gene duplication and evolution in recurring polyploidization – diploidization cycle in plants, doi: 10.1186/s13059-019-1650-2.
Response 2. Thanks for recommending these two excellent articles. I believe these two references are better cited with Reference 43 in the original submission. I have revised accordingly. The two recommended references are listed as No. 44 and No. 45 in the revision. (Line 100)
Comment 3. Line 111: What kind of epigenetic mechanism is involved in transcriptional silencing?
Response 3. The epigenetic regulation includes RNA-directed DNA methylation and histone H3K27 trimethylation that promote transcriptional silencing. I have added this sentence in the revised manuscript (Line 113).
Comment 4. Line 147: I did not find the table S1.
Response 4. I am sorry about this confusion. Table S1 was uploaded as part of the compressed file, named “manuscript-supplementary.zip”, per the journal’s instruction.
Comment 5. Line 170: I will highlight that the correlation between FBoX genes abundance and genome size is probably derived by the polyploidy levels that occur in a plethora of species. For instance, the Poaceae family are generally polyploid species, and this could explain the abundant of FBoX loci in some species belong to this family.
Response 5. Thanks for this insight about the contribution of genome polyploidization in expanding the size of the FBX gene family. To highlight this role, I have revised the sentence, that now reads, “Therefore, the birth and death of an FBX locus is rapid in plants. One mechanism could be attributed to the high rate of polyploidization events [43, 45].” (Lines 173-174)
Comment 6. Line 185: Please, could you indicate which statistical method was used for data analysis?
Response 6. The correlation was calculated using Spearman’s rank-order test. The method is added in the revision. (Lines 192-193)
Comment 7. Line 233: Please, indicate the statistical test used for the analysis.
Response 7. I think I have stated the method for the statistical test here. “Spearman’s r” indicates that the correlation is calculated using Spearman’s rank-order correlation test. (Line 242)
Comment 8. Line 269: based on the results, it emerges that only the Cluster 1 and cluster 2 presents a cnt value statistically significant compared with the other clusters. It should be useful to add more details to support your outcomes.
Response 8. I am not quite sure what this reviewer meant in this comment. The corresponding paragraph is to compare the lineage-specific duplications of the FBX subfamilies, exemplified by those from Cluster 1 in 111 plant genomes, with the large group of lineage-specific gene families in angiosperm. Fishers exact test demonstrated that the number of FBX subfamilies is over-represented in the lineage-specific group, suggesting that a significant proportion of lineage-specific FBX genes might not be functionally relevant but rather a consequence of a strong selection against their fixation in many genomes. To make this statement clearly, I added one sentence following Line 278. The new sentence states, “Therefore, the birth of many lineage-specific FBX genes is not necessary to be functionally relevant but rather a consequence of a strong selection against their fixation in many genomes.” (Lines 278-280)
Comment 9. Line 400: Please, could you add more details? For instance, the Poaceae families probably have much more genes in cluster 1 due to gene copy number variation induced by gene duplication?
Response 9. Thanks for this excellent point. The over-representation of Poaceae FBX genes in Cluster 1 group could be partially explained by yet unidentified Poaceae-specific genome duplication and fixation processes, which suggests a diversifying evolutionary process between Poaceae plants and the rest plant genomes analyzed. This unique duplication feature of Poaceae FBX genes was thoroughly discussed in Discussion (Lines 564-581).
Comment 10. Line 477: It is very interesting the CAF activity that you suggested. If it were possible, could you add more details? For instance, its binds to ASK1 in promotor region or in gene body?
Response 10. Thanks for the comment. However, I disagree with this reviewer’s view about the interaction of CAFs with the promotor region or the gene body of ASK1. Please note, an FBX protein physically interacts with the ASK1 protein to form an FBX-ASK1 heterodimeric protein complex that further assemble an SCF complex with Cul1 and RBX1. Their interactions are nothing to do with the promoter region or gene body of ASK1.
Comment 11. Line 480: I did not find the Figure 14.
Response 11. Thanks for the carefully reading. It is Figure S14 that was uploaded as part of the compressed file, named “manuscript-supplementary.zip”, per the journal’s instruction.
Comment 12. Line 489: I did not find the Table 2.
Response 12. Thanks for the carefully reading. It is Table S2 that was uploaded as part of the compressed file, named “manuscript-supplementary.zip”, per the journal’s instruction.
Comment 13. 458: I did not find Figure 12.
Response 13. Thanks for the carefully reading. It is Figure S12 that was uploaded as part of the compressed file, named “manuscript-supplementary.zip”, per the journal’s instruction.
Comment 14. Line 478: I did not find the Figure 13.
Response 14. Thanks for the carefully reading. It is Figure S13 that was uploaded as part of the compressed file, named “manuscript-supplementary.zip”, per the journal’s instruction.
Comment 15. Line 810: I did not find Figure 10.
Response 15. Thanks for the carefully reading. It is Figure S10 that was uploaded as part of the compressed file, named “manuscript-supplementary.zip”, per the journal’s instruction.